# Addressing Social Sustainability in Urban Regeneration Processes. An Application of the Social Multi-Criteria Evaluation

**Bottero Marta** and **Datola Giulia** *

Interuniversity Department of Urban and Regional Studies and Planning, Polytechnic of Turin, Viale Mattioli 39, 10125 Turin, Italy; marta.bottero@polito.it
* Correspondence: giulia.datola@polito.it

**Abstract:** The concept of sustainability is widely seen as fundamental to set up urban and territorial transformations. Sustainable development is a multidimensional and multi-perspective process that deals with the environmental, economic, and social dimensions, with the aim to find a balance among these. Despite this growing attention to sustainability the social perspective has been the less explored of these dimensions and only recently it is receiving consideration due the Sustainable Development Goals (SDGs) that aim at creating sustainable and inclusive cities and communities. In the SDGs, specific attention is focused on the improvement of the quality of life of inhabitants through specific actions dedicated to the valorization of cultural resources, to the protection of the environment, and also to promote the involvement of the local communities in setting policies and programs. The final objective is defining projects based on the social needs shared by the communities. This paper aims at exploring the social sustainability related to urban regeneration processes with particular attention to social cohesion and community engagement. Six different urban regeneration strategies, developed for the regeneration of an urban area located in Northern Italy and based on social housing interventions, have been evaluated in accordance with their social impacts on the stakeholders involved. The paper proposes a multi-methodological approach based on the combination of the stakeholder analysis with the NAIADE (Novel Approach to Imprecise Assessment and Decision Environments) methodology, a particular type of Social Multi-Criteria Evaluation. The stakeholder analysis has been applied to identify the actors to involve in the evaluation, whereas the NAIADE methodology has been implemented for the selection of the most preferable strategy. This method allowed the assessment of the different strategies through the comparison and the mediation between the technical and the social rankings, thus considering the stakeholder preferences in the final evaluation. The final result is coherent with the initial purpose and it demonstrates that the inclusion of the stakeholder is fundamental for the achievement of a consensus solution.

**Keywords:** social sustainability; multi-criteria analysis; urban regeneration; stakeholder analysis; NAIADE method

---

## 1. Introduction

During this last decade, social sustainability has been recognized as a fundamental component of sustainable development. This increasing attention is also recognized in the European policies and in the Sustainable Development Goals (SDGs). In detail, the present paper is focused on the social issues that are examined in specific goals, such as (1) increasing wellbeing (SDG 3), (2) reducing inequalities (SDG 10), creating resilient, inclusive, and safe cities (SDG 11), and promoting peaceful and inclusive societies (SDG 16) [1–3].

However, it has been widely recognized that the different dimensions of sustainable development (e.g., social, economic, environmental, and institutional) are not being equally prioritized by policy-makers within the sustainability discourse [4]. In fact, despite the abundance of social studies and policy documents, researchers have rarely approached sustainable development including equity and community engagement in the process.

In the literature, there is a relatively limited number of studies that focus specifically on social sustainability within its assessment, despite its recently increasing importance in setting urban and territorial transformations [1,5]. What clearly emerged from an in-depth literature review is that the concept of social sustainability is underdeveloped and often simplified in the existing theoretical frameworks [1,6,7]. Instead, social sustainability is a multidimensional concept. It deals with several social issues, such as inequality, displacement, and poor quality of livability [8–10]. Nowadays, there is a theoretical debate about both the meaning and the definition to use for rigorously addressing social sustainability. In fact, this concept includes different issues that belong to the philosophical, political, and practical fields. Therefore, it is complicated to determine its boundaries and define precisely what social sustainability means [1]. During the last decade, different scholars have observed social sustainability from different perspectives [11,12]. Some authors discuss about social sustainability in relation to democracy and equity [7], whereas others highlight the relationship between urban development and social sustainability focusing on community participation and engagement [9], also exploring the social dimension of sustainability through social impacts of physical elements and urban transformation [10,13,14]. In this context, different social sustainability definitions have been developed, and as a consequence, a wide range of approaches and methods for its assessment have been proposed. As an example, [15] identified at least 27 sustainability assessment techniques that have recently emerged in the literature and which are distinguished by different theories. Based on these circumstances, a comprehensive definition of social sustainability with a special focus on urban environments, provided by [16], has been chosen for this application. The final aim of this definition was putting the urban sustainability debate in relation with the physical environment (e.g., housing, urban design, public spaces) and its transformation, to assess the social impacts on the community involved in the regeneration process [17].

Considering both the necessity of a cross-disciplinary approach to analyze and assess social sustainability and the absence of consensus on which method to apply [10,18,19], this paper proposes the application of an integrated method based on the Social Multi-Criteria Analysis [20]. In particular, the NAIADE (Novel Approach to Imprecise Assessment and Decision Environments) method has been applied to perform and combine the technical rank and the social evaluation to assess the best alternative, considering for the evaluation the social impacts on the stakeholders.

The paper is structured as follows: Section 2 describes and compares the main approaches used to assess social sustainability; Section 3 is focused on the description of the NAIADE methodology to summarize its main characteristics; Section 4 is related to the presentation of the real case study and to the illustration of the evaluation process; Section 5 includes some final remarks and the future perspective.

## 2. Social Sustainability Assessment

As mentioned in the previous part, no consensus has been recognized in defining social sustainability. Therefore, several methods have been developed and adapted from different fields to evaluate social sustainability. This section describes and compares five of the main methods collected in the literature within their general frameworks, as shown in Table 1.

### 2.1. Social Return on Investment (SROI)

The Social Return on Investment has become one of the most applied approaches for assessing social impacts [21,22]. The SROI methodology was developed in 1996 by REDF (Roberts Enterprise Development Fund). It aims at evaluating the changes that certain projects can produce, in terms of

social, environmental, and the economic outcomes in monetary terms [23,24]. The evaluation is based on the assumption that each investment should consider both the financial value and the generated benefits. The final aim of the SROI method is determining the social values that are generated by an activity or organization.

The implementation of the SROI method within the context of urban projects is very recent and it is grounded on the Cost Benefit Analysis (CBA), putting more attention on the identification of the stakeholders involved in the process than the CBA.

From the methodological point of view, the SROI evaluation can be processed following these six phases [25]:

(1) Establishing the scope and identifying the stakeholders;
(2) Mapping the outcomes;
(3) Demonstrating the outcomes and giving them specific value;
(4) Establishing impacts;
(5) Calculating the SROI and performing the sensitivity analysis;
(6) Reporting.

### 2.2. Social Impact Assessment (SIA)

The Social Impact Assessment (SIA) is built on the principles of the Environmental Impact Assessment (EIA) [26–29]. The SIA method is addressed for managing and analyzing the social issues that can occur during planned policies and actions [30]. Therefore, the SIA method is mainly focused on the identification of the consequences of the current or future actions. It has been introduced in the context of urban transformations in the 70s and, actually, SIA methods are used to assist decision making and prioritization of social investment by project proponents [31].

The general procedure to process the SIA evaluation can be summarized as follows: (1) creating a participatory process with the objective to facilitate community discussion about the future actions and their impacts; (2) gaining a good understanding of the communities and actors that are affected by the policy under examination; (3) identifying the real community needs; (4) scoping the key social issues; (5) collecting the baseline data; (6) forecasting the social changes that may result from the policy; (6) establishing the significance of the predicted changes and also determining how various groups and communities will respond; (7) examining the other options; (8) developing a monitoring plan [19].

Moreover, the identification of the stakeholders involved in the process is fundamental within the SIA implementation. In fact, the final aim of the SIA methodology is assessing the consequences of actions in terms of impacts on the actors involved [19].

Therefore, the most important characteristics of the SIA, that implies its implementation in urban planning plans, can be argued as follows:

(1) The final aim is the identification of the social impacts generated by an action on the community and on the citizens (the stakeholders involved);
(2) The results obtained by the SIA methodology are useful to support the decision making process of a transformation project, according to its social impacts;
(3) It is applied in the ex-ante phase, so it is suitable to evaluate in advance the social impacts, both positive and negative;
(4) It is able to increase the community consciousness about the intervention and its consequences.

### 2.3. Social Multi-Criteria Evaluation (SMCE)

The technique of the Social Multi-Criteria Evaluation (SMCE) has been developed by Munda with the aim of integrating the Multi-Criteria Analysis (MCA) with technical and social issues [20]. SMCE can be considered a specific typology of Multi-Criteria Analysis (MCA) that is focused on the social dimension of a problem. More in detail, SMCE is grounded on the principle of the necessity of

extending the MCA with the incorporation of the notion of the stakeholder. In fact, the stakeholder participation is used as the input of the analysis itself in the SMCE process [32]. Addressing the position and the role of the stakeholder is fundamental when dealing with complex systems in which the actors can have conflicting and legitimate opinions about the possible solutions of the problems. Based on these circumstances, the evaluation process related to this method has to be participative and transparent [20]. Furthermore, in SMCE, the participation is necessary but not sufficient [32] because the transparency plays a crucial role, allowing us to underline and express which are the values and which stakeholder groups are favored by each option.

Based on these characteristics, the SMCE aims at analyzing the decision making processes in complex and interdisciplinary perspectives, considering the plurality of objectives of different stakeholders involved.

The main principles on which the SMCE is grounded in can be summarized as follows:

(1)    Definition of the problem;
(2)    Institutional analysis;
(3)    Generation of the policy options;
(4)    Construction of the multi-criteria impact matrices;
(5)    Application of the mathematical procedure;
(6)    Sensitivity analysis.

## 2.4. Social Life Cycle Assessment (S-LCA)

The Social Life Cycle Assessment (S-LCA) is grounded on the Life Cycle Thinking (LCT) approach. In detail, it is one of the three techniques that compose the Life Cycle Sustainability Assessment (LCSA) [33–35], that allows the assessment of sustainability within its three different dimensions: (1) economy, (2) environment, and (3) society. Figure 1 illustrates the three different techniques that compose the Life Cycle Sustainability Assessment or rather: (1) Environmental Life Cycle Assessment (E-LCA), (2) Social Life Cycle Assessment (S-LCA), and (3) Life Cycle Costing (LCCA).

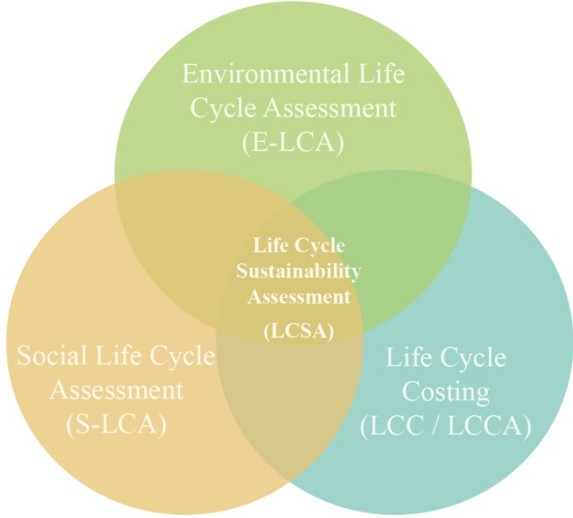

**Figure 1.** Life Cycle Sustainability Assessment (LCSA) (source: [36]).

Social Life Cycle Assessment (S-LCA) allows the evaluation of the social impacts of products and processes on the interested stakeholders. Its framework considers two categories: (1) stakeholder category and (2) impacts category [37,38]. Thus, it is possible to highlight the most significant social and socio-economic aspects within the life cycle of products/processes.

The evaluation is processed following these main steps:

(1) Definition of the evaluation objective. In detail, the aim of the S-LCA implementation is strictly related to the use of the product under examination.

(2) Inventory analysis. This phase concerns the collection of data and information, useful to develop the analysis. The inventory phase foresees the identification of the indicators to use for the evaluation of the impacts.

(3) Evaluation of the impacts. This phase is dedicated to the assessment of the product's impacts on the stakeholders identified.

(4) Results explanation. The last phase is dedicated to the interpretation of the obtained results, in order to develop a final report in which the involvement of the stakeholders is described.

## 2.5. Community Impact Evaluation (CIE)

The Community Impact Evaluation (CIE) is a multi-actor evaluation methodology. This method has been developed to respond to the weaknesses of the traditional evaluation methods, such as the Cost Benefit Analysis (CBA) [39]. The CIE aims at identifying the convenience of projects according to the social preferences expressed by the stakeholders involved [40]. Therefore, the implementation of the CIE methodology has become relevant to support the decision making in urban regeneration and transformation processes [40–42]. In detail, the CIE evaluation steps can be summarized as follows [42]:

(1) Characterization of the project. In the first phase, the project has to be described in-depth, providing also information of the context in which it will be located;

(2) Mapping the stakeholder. The second phase concerns the identification and the mapping of the social groups interested by the project. The mapping is based on their spatial location (on site or off site) and over time (in short and medium-long term). Moreover, as suggested by [43], the stakeholders have to be categorized into two macro-groups. The first group represents the active stakeholders, such as operators and producers. The second group illustrates passive actors, such as the consumers who use goods and services;

(3) Analysis. This phase is structured into two subsequent steps. The first one is defining the project's objectives, through which it will be evaluated in terms of impacts on stakeholders. The second step concerns the identification of the effects for the groups of interest;

(4) Descriptive assessment. In this phase, the impacts are evaluated both in a qualitative and quantitative way. Specifically, the final evaluation is performed through a final intersection grid that summarizes the social preferences of stakeholders with the impacts of the project.

## 2.6. Overview of Methods for Social Sustainability Assessment

Table 1 compares the five methodologies above illustrated, highlighting (1) the evaluation objective, (2) the derivation method, (3) the presence of the monetization of social benefits, (4) the typology of the evaluation, (5) the participation role, and (6) the application of the methods in urban or territorial fields.

Table 1. Comparison of the described methods (elaboration from [36]).

| | Evaluation Objective | Derivation Method | Monetization of Social Benefits | Typology of Evaluation | Participation Role | Application in Urban or Territorial Field |
|---|---|---|---|---|---|---|
| SROI | Social impacts, and socio-economic impacts | [Social Balance BS + CBA] | Yes | Ex-ante Ex-post | Necessary | Urban regeneration policies [44]; Social Housing [24]; Rural development in England [45] |
| SIA | Social impacts, and socio-economic impacts | [EIA] | No | Ex-ante | Necessary | Land requisition [46]; Rebuilding a neighborhood [47]; urban regeneration [48] |
| SMCE | Social impacts | [MCA] | No | Ex-ante | Necessary, but not sufficient | Urban sustainability policies [49]; Windfarm location [50] |
| S-LCA | Social impacts | [LCA + LCC] | No | Ex-ante | Necessary | Not actually (the principle of Life Cycle Thinking is actually applied to evaluate a single sector of an urban system) [51] |
| CIE | Social impacts | [CBA] | No | Ex-ante | Necessary | Urban regeneration process [40]; Urban restoration [41]; Smart city [42] |

## 3. Method

The present paper proposes the multi-methodological approach based on the combination of the stakeholder analysis with the NAIADE methodology to analyze six different urban regeneration strategies. This section aims at briefly describing these two techniques within their main characteristics.

### 3.1. Stakeholder Analysis

The Stakeholder Analysis (SA) is a technique used to define strategies through the identification of the key actors within their objectives and interests [52]. In detail, identifying and analyzing the interest of the different stakeholders is fundamental within urban regeneration processes [53,54]. Thus, it is possible to identify in advance possible conflicts among them and also to better recognize their needs and requirements [54]. From the practical point of view, stakeholders are classified according to their objectives and to the resources that they can carry out in the process (i.e., political, economic, legal, and cognitive resources) [52]. Therefore, it is possible to divide stakeholders into five categories, namely political, bureaucratic, special interest, general interest, and experts. Different methodologies can be applied to map stakeholders and actors, such as the Power/Interest Matrix [55], the Stakeholder Circle Methodology [56] and the Social Network Analysis [53,54,57].

In detail, in this paper, the Stakeholder Circle Methodology is applied to map the stakeholders involved (Section 5.1). This specific technique, developed by Bourne [56] analyzes and maps the stakeholders according to their proximity, power, and interest. Moreover, it permits in this application to list the stakeholders according to these three criteria to determine which are the key players in the process.

### 3.2. NAIADE Methodology

The NAIADE methodology (Novel Approach to Imprecise Assessment and Decision Environments) refers to the Multi-Criteria Analysis (MCA). It belongs to the Social Multi-Criteria Evaluation approach, developed by Munda [20,32,50,58] as a framework to apply social choice in complex political problems to focus on the stakeholders and their specific interests. Considering the peculiarities of the SMCE (Section 2.3), the NAIADE method has been widely applied in many different

fields, and also in urban and environmental contexts. Table 2 summarizes the main application of the NAIADE method in urban and environmental fields.

**Table 2.** Literature review on the NAIADE approach in the context of urban and territorial transformation projects (elaboration of [59]).

| Author and Year | Decision Problem Context | Journal |
|---|---|---|
| Crescenzo et al., 2018 [60] | Urban planning | Green Energy and Technology |
| Nicolini and Pinto, 2013 [61] | Urban planning | Sustainability |
| Garmendia and Gamboa, 2012 [62] | Natural resource management | Ecological Economics |
| Monterroso et al., 2011 [63] | Ecosystem management | Journal of Environmental Management |
| Oikonomou et al., 2011 [64] | Protected area management | Environmental Management |
| Garmendia et al., 2010 [65] | Integrated coastal zone management | Ocean and Costal Management |
| Shmelev and Rodriguez-Labajos, 2009 [66] | Sustainability assessment | Ecological Economics |
| Ramírez et al., 2009 [67] | Environmental management | Energy Procedia |
| Gamboa, 2006 [68] | Environmental management | Ecological Economic |
| Munda, 2006 [58] | Sustainability assessment | International Journal of Environmental technology and management |
| Sturiale and Scuderi, 2019 [69] | Green infrastructure and climate change | Climate |
| Della Spina, 2019 [70] | Urban regeneration | Sustainability |
| Stanganelli et al., 2019 [71] | Urban regeneration | Sustainable cities and society |

The peculiarity of the NAIADE method stands in the development of two different types of evaluations, that are:

(1) The technical evaluation. It is grounded on the score assigned to the criteria of each alternative and it is performed using an impact matrix (alternatives vs. criteria). In this case, the final output given by the NAIADE method is represented by the ranking of the alternatives, processed in accordance to the set of criteria preferences;

(2) The social evaluation that explores the conflicts among the different stakeholders. Furthermore, through this evaluation it is possible to explore the probable coalitions among different stakeholders using an equity matrix, which provides a linguistic evaluation of alternatives by each group.

Moreover, this methodology is structured to include both the qualitative and quantitative variables in the evaluation. The different typologies of variables that NAIADE is able to include can be summarized as follows:

(1) *Crips*, which values can be defined between only two different options;

(2) *Fuzzy*, that represent those variables defined as "uncertain" or "blur", for which infinite values can be assigned;

(3) *Stochastic* or rather "casual" because their values can vary continuously.

## 4. Case Study

The proposed multi-methodological approach is applied to evaluate six urban regeneration strategies, in accordance with their social impacts on the stakeholders involved. Specifically, these actions

have been developed for the regeneration program "Collegno Rigenera" for the city of Collegno (Northern Italy). This program has been promoted by the municipal administration and it is focused on the requalification of a specific area of the municipal territory that is characterized by economic and social fragility. The main challenge of this program is finding answers to the economic and social needs [72].

In the present case study, an integrated approach based on (1) stakeholder analysis and (2) the NAIADE methodology has been implemented to address the complexity of the decision problem under examination.

*Urban Regeneration Strategies*

As mentioned before, in this application the NAIADE method has been applied to evaluate the social impacts of six different regeneration strategies on the stakeholders involved in the process. The developed scenarios can be described as follows:

(1) Cultural District. This strategy aims at creating both social housing to respond to the necessity of the university students and at realizing cultural activities for the area, including a new library for residents and students;

(2) Smart City. The goal of this project is trying to give to the area a new identity. The major intervention is the creation of social housing blocks adapted to students, families, and the elderly;

(3) Start Up. This project is focused on the creation of social housing mixed with new activities, in order to improve both the social and the economic conditions of the area;

(4) City and Craft. This strategy is mainly focused on the valorization of the economic activities. In fact, in this project the realization of a new social housing block aims at revitalizing the area in order to attract also new economic activities;

(5) Sharing City. The main objective of this strategy is the creation of the common spaces to implement the community engagement and cohesion. Due to this, the social housing blocks foresee different common spaces;

(6) Green Infrastructure. This strategy aims at integrating new constructions with green spaces. In fact, the new housing blocks are connected with each other through green corridors and pedestrian paths.

## 5. Application

### 5.1. Stakeholders Involved in the Process

Before applying the NAIADE methodology, the stakeholders analysis has been performed to identify the stakeholders influenced by the urban regeneration process and to determine their objectives, interests, and resources. As mentioned before, this paper applies the Stakeholder Circle Methodology because it is able both to analyze and to map the actors involved, focusing on their power, and their proximity and urgency in the process, starting from their characteristics [52].

Table 3 surveys the stakeholders involved in the transformation process, with a specific reference to the level, the type, the resources, and the goal that they follow within the process.

**Table 3.** Survey of the stakeholders involved in the process (source: [36]).

| Stakeholder | Level | Category | Resources | Objective |
|---|---|---|---|---|
| European Union | European | Political | Political | Political consensus |
| Piedmont Region | Regional | Political | Political | Improvement of the condition of the regional territory and political consensus |
| Metropolitan city of Turin | Local | Political | Political | Creation of the network between the different municipalities |
| Collegno Municipality | Local | Political | Political | Improvement of the social, economic, and urban conditions through the implementation of the regeneration process |
| Municipality technical office | Local | Bureaucratic | Legal | Improvement and protection of the municipal territory |
| Developer | Local | Special interest | Economic | Maximize the economic income |
| Business owners | Local | Special interest | Economic | Improving the condition of the area in which their activities are located to increase their economic incomes |
| Land owners | Local | Special interest | Economic | Maximize their economic income related to the increasing of the value of their properties |
| Sponsors | Regional | Special interest | Economic | Improving their visibility through the participation at the urban regeneration program |
| Associations | Local | General interest | Cognitive | Achievement of the social wellbeing and protection of the environmental and historic capital |
| Residents | Local | Special interest | Cognitive | Improvement of both residential and employment conditions in order to get better community cohesion |
| Students | Local | Specific interest | Cognitive | Increasing the studying services |
| Tourists | Local | Specific interest | Cognitive | Having new cultural attractions |
| Planners | Local | Experts | Cognitive | Economic income |
| Technicians | Local | Experts | Cognitive | Economic income |
| Media | Local | General interest | Cognitive | Exchange about territory information |
| Transportation Society (GTT) | Regional | Specific interest | Cognitive | Improvement of the transportation service |
| Artisans | Local | Specific interest | Cognitive | Improvement of the connection of this area with the other municipalities to increase the commercial opportunities |

Figure 2 illustrates the result of stakeholder analysis performed through the Circle Methodology. As shown in Figure 2, stakeholders have been mapped considering (1) their power, that is represented by the dimension of the wedge they occupy, (2) their proximity, that is figured out by the concentric circles, and (3) their urgency, that is illustrated by the depth of the wedge. Through this analysis it was possible to determine the role and the position of different stakeholders in reference to the urban regeneration process. In detail, the developer, the technical office, and the municipality of Collegno can be considered key players within the urban regeneration process. Therefore, their power and proximity are relevant, and their urgency can reach the goal. Instead, land and building owners, business owners, inhabitants, planners, and technicians have medium power and high proximity and urgency. Thus,

the analysis has been fundamental to clarify the most relevant stakeholders to include in the social evaluation performed with the NAIADE method.

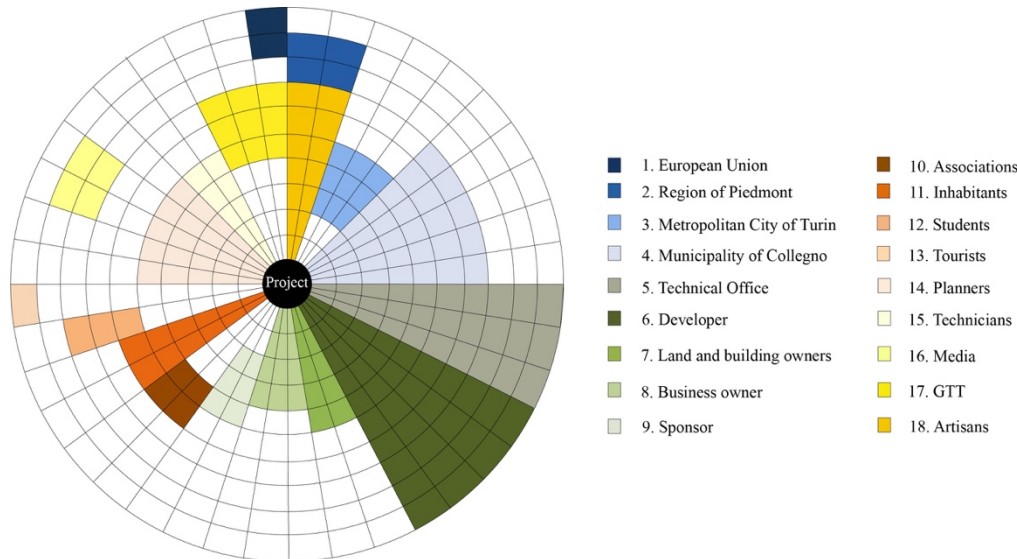

**Figure 2.** Stakeholder circle analysis (source: [36]).

## *5.2. Development of the NAIADE Methodology*

### 5.2.1. Identification of the Criteria

The first step for the application of the NAIADE methodology concerned the identification of the criteria to use to evaluate the performance of each alternative. Table 4 lists the criteria considered in this application that are divided into five categories, namely, (1) sharing, (2) environment, (3) service, (4) mobility and accessibility, (5) economy, and (6) regeneration. In detail, these criteria have been selected during a focus group with experts and stakeholders [72]. Thus, it was possible to recognize their interests and objectives in the evaluation.

**Table 4.** List of criteria used for the evaluation [72].

| Criteria Category | n. | Criterion | Unit | Description |
|---|---|---|---|---|
| Sharing | 1 | Public space/private space | [-] | Ratio between public and private surfaces |
| | 2 | Co-working space | [m$^2$] | Surface of the structures for workshops, meetings, and training courses |
| | 3 | Co-housing inhabitants | [num.] | Number of residents in new co-housing buildings |
| Environment | 4 | Permeable surf./Territorial surf. | [-] | Ratio between permeable areas and overall territorial surface of the program |
| | 5 | Urban gardens | [m$^2$] | Total area used for community and private urban gardens |
| | 6 | Waste production | [kg/year] | Amount of waste produced in a year by the activities of the program |

**Table 4.** *Cont.*

| Criteria Category | n. | Criterion | Unit | Description |
|---|---|---|---|---|
| Services | 7 | Residence | [m$^2$] | Surface for residential functions |
| | 8 | Commercial areas | [m$^2$] | Surface for commercial functions |
| | 9 | Sports and cultural areas | [m$^2$] | Surface for sport and cultural activities |
| | 10 | Mixité index | [0–1] | Index that describes the functional mix of the area |
| Mobility/Accessibility | 11 | Slow mobility | [m$^2$] | Surface of the pedestrian tracks and bicycle lanes |
| | 12 | Car parking | [num.] | Number of new public parking lots |
| | 13 | Bike or car sharing points | [num.] | Number of car and bike sharing points |
| Economy | 14 | Total Economic Value | [€] | Estimate of the social benefits delivered by the program |
| | 15 | Investment cost | [€] | Total cost of the program |
| | 16 | New jobs | [num.] | Number of new jobs created |
| Regeneration | 17 | Regeneration | [m$^2$] | Regenerated surface |
| | 18 | Via De Amicis regeneration | [qualitative scale] | Qualitative index showing the level of the regeneration of Via De Amicis |
| | 19 | Territorial Index | [-] | Ratio between the maximum buildable volume and the territorial surface |

### 5.2.2. Technical Evaluation: Impact Matrix

Once the criteria to use were identified, the first step of the application of the NAIADE methodology was the development of the impact matrix (Appendix A). It evaluates the different scenarios according to the set of multidimensional criteria (both qualitative and quantitative) that includes all the relevant aspects of the decision problem. From this evaluation, a first technical ranking has been obtained, as shown in Figure 3.

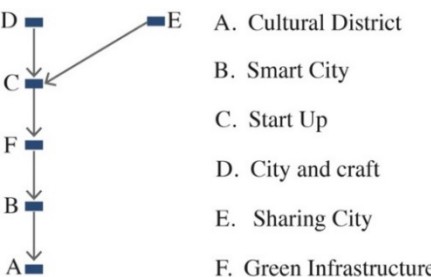

**Figure 3.** Technical ranking.

Resulting from the technical ranking, the two most preferred solutions are Scenario D, City and Craft, and Scenario E, Sharing City. In fact, these two scenarios have good performances (Appendix A) in the majority of the considered evaluation criteria. As an example, both scenarios have a very good performance in the criterion "total economic value" that has been applied to monetize the social benefits of the interventions. Moreover, Scenario D, City and Craft, gives great importance to the

criterion "urban gardens" that is considered fundamental by the stakeholders involved. Whereas the Sharing City scenario assigns a great relevance to the criterion "sport and cultural area" that is one of the main points of the "Collegno Rigenera", in order to make this area inclusive.

### 5.2.3. Social Evaluation: Equity Matrix

According to the NAIADE approach, a second matrix has been defined that is the equity matrix, as shown in Table 5. This matrix illustrates the assessment of each scenario, expressed in a qualitative scale by each stakeholder involved in the evaluation. Differently from the impact matrix, in the equity matrix stakeholders are allowed to evaluate each alternative using linguistic variables. In detail, the evaluation is processed by the analyst that examines the stakeholders' opinions, combining also the stakeholder analysis. Specifically, in this application, a multi-level scale has been considered to implement this matrix. Following the NAIADE methodology [20], the considered scale is composed of nine qualitative points that are (1) perfect, (2) very good, (3) good, (4) more or less good, (5) moderate, (6) more or less bad, (7) bad, (8) very bad, and (9) extremely bad. From this matrix, it is possible to examine the distributional issues. Specifically, using a distance function $d_{ij}$ as a conflict indicator, a similarity matrix $s_{ij} = 1/(1 + d_{ij})$ can be constructed for all possible pairs of groups, so that a clustering procedure is meaningful. By applying this procedure to the social impact matrix, a coalition dendrogram can be obtained, as shown in Figure 4.

**Table 5.** Social impact matrix.

| | Alternatives | | | | | |
|---|---|---|---|---|---|---|
| | **Cultural District** | **Smart City** | **Start Up** | **City and Craft** | **Sharing City** | **Green Infrastructure** |
| Developer G1 | Moderate | More or less bad | Very Good | More or less bad | Very bad | Moderate |
| Municipality G2 | Good | More or less bad | More or less good | Good | Good | Very good |
| Technical Office G3 | Good | Moderate | Moderate | Good | More or less good | More or less bad |
| Planners G4 | More or less good | Moderate | Moderate | Good | More or less good | Moderate |
| Artisans G5 | Good | Good | Very Good | Perfect | More or less good | More or less good |
| Land and Building Owners G6 | More or less good | More or less bad | Moderate | Moderate | Good | Very good |
| Inhabitants G7 | More or less good | More or less bad | Moderate | Moderate | Good | Very good |
| Business Owners G8 | Moderate | Very good | Perfect | Good | Good | More or less good |

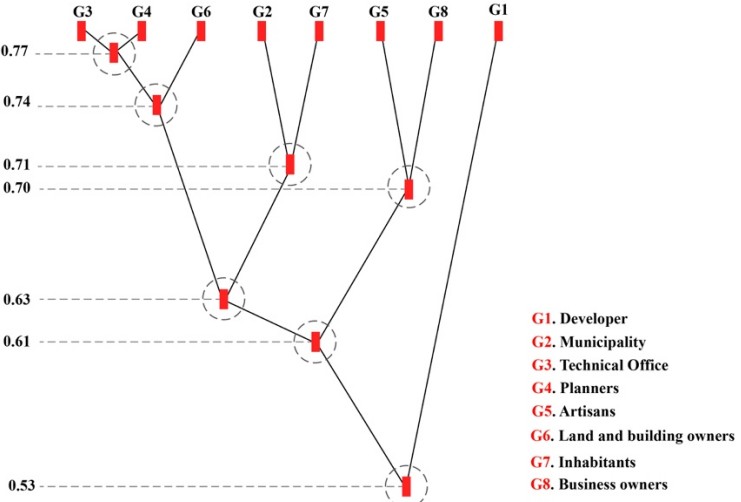

**Figure 4.** Dendrogram of coalitions.

## 6. Discussion of the Results

Figure 4 shows the dendrogram, through which it is possible to visualize the proximity of the stakeholders involved. The first coalition is built by the Technical Office (G3) and Planners (G4), and their proximity is very high (0.77) because both pursue the objective of the requalification of the area. Secondly, the abovementioned coalition is joined by Land and Business owners (G6), with a very high credibility (0.74). This can be justified by the fact that these three stakeholders aim at reaching the development and the improvement of the transformation area. Another coalition with a great credibility (0.71) is performed by the Municipality (G2) and Inhabitants (G7). In fact, both the stakeholders aim at improving the social and economic condition of this area. Thirdly, also the coalition between Artisans (G5) and Business owners (G8) has a great proximity (0.70). This is due to the fact that both Artisans and Business owners can have economic benefits from the improvement of the social conditions of the area. Moreover, some other coalitions with medium proximity have been identified. The first one, with the proximity of 0.63, is shaped by the joint between Technical Office (G3), Planners (G4), and Land and Buildings owners (G6), with Municipality (G2) and Inhabitants (G7). The second, with 0.61 of credibility, is the result of the joint between the abovementioned coalition (G3 + G4 + G6 + G2 + G7) with Artisans (G5) and Business owners (G8). The last coalition with medium-low proximity (0.53) is shaped by the combination of a coalition (G3 + G4 + G6 + G2 + G7 + G5) with the Developer (G1). This is interesting because it allows us to underline that the Developer has a very different objective from the other stakeholders. It was also possible to underline their interest in the economic return of the investment [73].

As suggested by [50], it is also important to combine the analysis of the social impact matrix (Table 5) with the dendrogram to give a robust interpretation of the obtained results to the decision makers. In this sense, it is possible to highlight that for the Technical Office (G3) and Planners (G4), the best solution is the alternative City and Craft, followed by the alternative Cultural District. Instead, for the Land Owners (G6), the preferred alternatives are the Sharing City and Green Infrastructure; however, also the scenario City and Craft is moderately good. Considering the coalition (G3, G4, and G6), the preferred solutions are the scenarios Cultural District and City and Craft. The Municipality (G2) and the Inhabitants (G7) are in accordance in considering the Smart City scenario as the worst alternative, whereas they consider good/more or less good the scenarios City and Craft, Sharing City, and Green Infrastructure. Finally, Business Owners (G8) and Artisans (G5) agree in appreciating the scenario City and Craft. Whereas the Developer (G1) prefers the Start Up scenario.

Considering that the main aim of this evaluation was assessing the different regeneration strategies considering both their social impacts and their technical performance, this application develops a

comparison and mediation between these, obtaining a multi-ranking evaluation. Figure 5 illustrates the comparison. In detail, the social ranking has been performed considering both the social impact matrix, that is shown in Table 5, and the dendrogram (Figure 4). Thus, it was possible to interpret and visualize the ranking of the alternatives according to the preferences expressed by the involved stakeholders. From the technical rank, the best performing scenarios are "City and Craft" and "Sharing City", as shown in Figure 5, while from the social point of view, the preferable strategy seems to be the "City and Craft" scenario. According to the results of the evaluation, the preferable scenario is "City and Craft", because it can combine both the technical and the social performances in order to maximize both the technical and the social impacts.

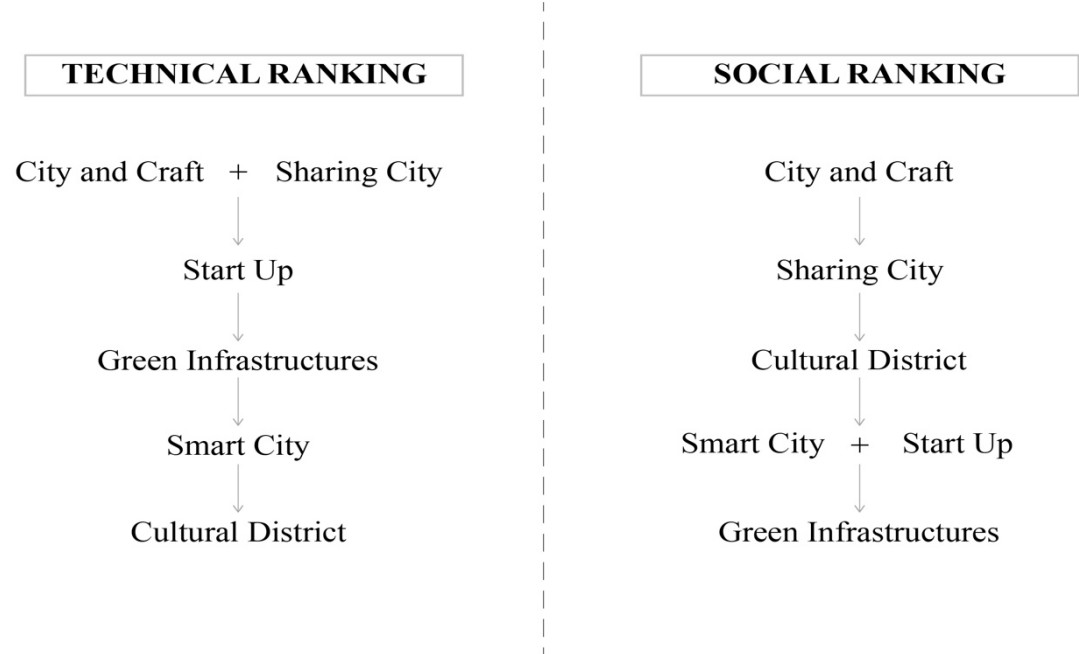

**Figure 5.** Comparison between technical and social ranking.

## 7. Conclusions

The present research proposes an investigation of the evaluation methods for addressing social sustainability within urban regeneration processes. In particular, the study illustrates the application of the NAIADE method combined with the stakeholder analysis for assessing different urban regeneration strategies, according to their social impacts on the stakeholders involved. In this decision process, characterized by a high level of complexity and different legitimate values and objectives, this method allowed the consideration of both the technical criteria and the opinions of the stakeholders involved [52,54]. This application highlights the importance of the involvement of the stakeholders within the evaluation process. Thus, it was possible to determine to which social impacts the stakeholders are exposed. Furthermore, it also underlines that the participation of the stakeholders is a necessary requirement to obtain social sustainability and to promote a consensus solution in the urban regeneration process [74–77]. Moreover, the results obtained by the social evaluation processed with NAIADE are comparable with the results obtained with other evaluation methods [72]. In fact, in these different evaluations, the most preferable scenario is the Sharing City. Thus, it can be demonstrated that also the social evaluation is fundamental in supporting urban decision processes, giving robust recommendations.

The main strength of using the NAIADE method for our purpose is represented by the social impact matrix and coalition dendrogram. In fact, in the equity matrix, the alternatives have been evaluated considering the social impacts on the same stakeholder, while the dendrogram shows the

coalition from a social point of view. The results obtained are highly coherent and the approach has proven strength. Furthermore, this application also demonstrates the suitability of using the NAIADE method to assess social sustainability, focusing on its relationships with the urban environment and its transformation [49,50,62], The application presented in the paper has allowed us to underline also the weaknesses of this method, or rather the method through which the social matrix and the comparison of rankings are performed. For this reason, future research and applications can be addressed to find a method to perform the social matrix and the implementation of the combination of the two different rankings in a more rigorous way, since it is actually performed in a qualitative way. Finally, further development could also consider the performing of a specific sensitivity analysis to better verify the model with the perspective to formulate more robust recommendations.

**Author Contributions:** All authors contributed equally to the development of this paper: Conceptualization, B.M., D.G.; Investigation, B.M., D.G.; Validation, B.M., D.G.; Writing—original draft preparation, B.M., D.G.; Writing—review and editing, B.M., D.G. All authors have read and agreed to the published version of the manuscript.

**Funding:** This research received no external funding.

**Acknowledgments:** The authors wish to thank Angela Dho for the data used in the present study.

**Conflicts of Interest:** The authors declare no conflict of interest.

# Appendix A

**Table A1.** Impact matrix (elaboration from Bottero et al., 2017).

| Criteria Category | n. | Criterion | Units | Scenarios | | | | | |
|---|---|---|---|---|---|---|---|---|---|
| | | | | Cultural District | Smart City | Start Up | City and Craft | Sharing City | Green Infrastructure |
| Sharing | 1 | Public space/private space | [-] | 4.31 | 3.25 | 1.33 | 8.35 | 2.76 | 4.20 |
| | 2 | Co-working space | [m$^2$] | 20,425 | 24,260 | 49,880 | 11,328 | 5108 | 3300 |
| | 3 | Co-housing inhabitants | [num.] | 398 | 150 | 255 | 421 | 2513 | 1036 |
| Environment | 4 | Permeable surf./Territorial surf. | [-] | 0.69 | 0.39 | 0.58 | 0.52 | 0.53 | 0.71 |
| | 5 | Urban gardens | [m$^2$] | 8527 | 2130 | 25,569 | 66,894 | 23,118 | 12,888 |
| | 6 | Waste production | [kg/year] | 1,350,845 | 2,332,234 | 2,692,663 | 1,817,205 | 3,014,301 | 1,631,941 |
| Services | 7 | Residence | [m$^2$] | 70,880 | 117,736 | 82,330 | 164,925 | 538,018 | 75,252 |
| | 8 | Commercial areas | [m$^2$] | 28,031 | 59,169 | 95,000 | 84,248 | 40,192 | 25,515 |
| | 9 | Sports and cultural areas | [m$^2$] | 48,150 | 81,796 | 26,960 | 21,458 | 114,725 | 37,920 |
| | 10 | Mixitè index | [0–1] | 0.71 | 0.46 | 1 | 0.30 | 0.30 | 0.64 |
| Mobility/Accessibility | 11 | Slow mobility | [m$^2$] | 68,326 | 171,609 | 16,000 | 132,541 | 624,933 | 251,831 |
| | 12 | Car parking | [num.] | 1385 | 2567 | 2100 | 1137 | 1689 | 1394 |
| | 13 | Bike or car sharing points | [num.] | 7 | 12 | 2 | 3 | 14 | 19 |
| Economy | 14 | Total Economic Value | [€] | 2,550,746 | 537,692 | 3,500,000 | 7,471,328 | 7,707,778 | 531,155 |
| | 15 | Investment cost | [€] | 233,336,184 | 279,468,021 | 100,000,000 | 183,948,594 | 494,055,026 | 231,527,860 |
| | 16 | New jobs | [num.] | 1010 | 1545 | 300 | 736 | 3229 | 768 |
| Regeneration | 17 | Requalification index | [-] | 0.20 | 0.12 | 0.51 | 0.36 | 0.06 | 0.20 |
| | 18 | Via De Amicis Requalification | [qualitative] | Fair | Excellent | Good | Good | Very good | Very good |
| | 19 | Territorial Index | [m$^2$] | 0.38 | 0.16 | 0.23 | 0.52 | 0.40 | 0.13 |

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
