# Peer review of "Addressing Social Sustainability in Urban Regeneration Processes. An Application of the Social Multi-Criteria Evaluation"

_sustainability, doi:10.3390/su12187579_

Round 1
Reviewer 1 Report
General comments
The research question is original and well defined and concerns a very topical issue.
The results provide an advance in current knowledge and above all an interesting integration of actual themes in regeneration process. They meet the objective of giving solidity and empirical evidence to a topic addressed only in a theoretical way and in a still poorly defined way. The paper presents an interesting input and a very clear methodology for assessing social impacts on the stakeholder involved in different urban regeneration strategies. Given the complexity and variety of perspectives through which the issue of social sustainability can be addressed, the proposal for a multi-methodological approach was highly appreciated
The conclusions demonstrate the need to use an empirical evaluation approach to make participation in regeneration processes truly operational, realistically considering the social impacts on different stakeholders. The article is written using a simple language and the use of the English language is very appropriate and understandable.
The issues addressed in the paper are of great interest to readers from different scientific fields and represent an excellent starting point for reflection on how to include social needs in the definition of designing strategy.
The paper proposes work that can be further implemented but opens up interesting perspectives for the achievement of long-term goals.
Specific comments
It is better to specify the acronym of the NAIADE already in the abstract, in which it appears for the first time.
SDG’s are mentioned in the abstract but are not included in the text. It would be appropriate to mention them in the introductory part specifying how the issue of sustainability at social level is now also present in international documents, assuming the different perspectives that you have cited in the paper. (I would also point out some documents of the European Commission in which this attention is drawn to this issue. I would also suggest this European Parliament document: https://www.europarl.europa.eu/RegData/etudes/STUD/2020/648782/IPOL_STU(2020)648782_EN.pdf)
Author Response
Letter to Reviewer #1
Thank you a lot for appreciating our work. We very appreciate your comments that were useful to improve our work.
Moreover, we would like to underline that the general structure of the paper has been changed, in order to respond to all the suggestions received. However, the content of the paper is the same, considering that has been accepted by all the reviewers.
The authors.
Specific comments
It is better to specify the acronym of the NAIADE already in the abstract, in which it appears for the first time.
- Thank you for this suggestion, we added the description of the acronym also in the abstract.
SDG’s are mentioned in the abstract but are not included in the text. It would be appropriate to mention them in the introductory part specifying how the issue of sustainability at social level is now also present in international documents, assuming the different perspectives that you have cited in the paper. (I would also point out some documents of the European Commission in which this attention is drawn to this issue. I would also suggest this European Parliament document: https://www.europarl.europa.eu/RegData/etudes/STUD/20 20/648782/IPOL_STU(2020)648782_EN.pdf)
- Thank you for this suggestion, we very appreciated it. We inserted the reference to SDG in the introduction, underline which are the aspects of social sustainability we analyse in the paper.

Reviewer 2 Report
The paper proposes an investigation of the evaluation methods for addressing social sustainability within urban regeneration processes. The topic may still be of interest but the current manuscript needs a major revision, extension and is not in the current form ready for publication. Although, the work is also of interest, the structure followed is not clear and a number of issues must be clarified before a publication recommendation may be made.
The authors propose “a multi-methodological approach based on the combination of the stakeholder analysis with the NAIADE methodology”. However, they present the proposed method approach with minimum definition. Therefore, this paper needs a major revision and precisely editing to provide a more adequate description of the novelty of proposed multi- methodological approach. Moreover, the current form of this paper is not well organized and the paper is in mess.
The authors present a state-of-the-art of “Social Sustainability Assessment” and “NAIADE Methodology”. It is suggested that the methodology approach proposed by the authors be presented later in an independent section (Methods), prior to its application to the case study: Collegno. This section should be clear, so that the method can be replicated by another researcher in another case study.
Discussion of paper is poor and the analysis should be presented with more description. The main contribution of the authors in comparison with previous works should discuss more profoundly and highlights in detail. The authors should explain clearly, what the contribution of this paper is. It should not be a mere application of an already known methodology applied to a case study.
Conclusions. The paragraph shall be independent from the main text without citing references. It should be concise including the most important new findings. What are the limitations of the proposed method?
Other issues:
What information is obtained from each of the skateboarders should be provided. How are defined the matrices (impact and equity)?
Please, avoid lumping references as in [1-5], [26-29] and all other. Instead, summarize the main contribution of each referenced paper in a separate sentence.
The language should be revised as grammatical errors are pervasive: there are typos, wrong figure captions…
Author Response
Letter to Reviewer #2
We thank you for your comments and suggestions. They were very useful to improve the structure and the quality of our work.
Comments for authors
The paper proposes an investigation of the evaluation methods for addressing social sustainability within urban regeneration processes. The topic may still be of interest but the current manuscript needs a major revision, extension and is not in the current form ready for publication. Although, the work is also of interest, the structure followed is not clear and a number of issues must be clarified before a publication recommendation may be made.
The authors propose “a multi-methodological approach based on the combination of the stakeholder analysis with the NAIADE methodology”. However, they present the proposed method approach with minimum definition. Therefore, this paper needs a major revision and precisely editing to provide a more adequate description of the novelty of proposed multi- methodological approach. Moreover, the current form of this paper is not well organized and the paper is in mess.
- Thank you for these comments. We reviewed the general structure of the paper. We inserted the paragraph “method” in which we underlined that in this specific application we combined the stakeholder analysis and the NAIADE methodology. Moreover, we inserted a specific paragraph dedicated to the illustration of the main characteristics of the stakeholder analysis. We also added a specific paragraph dedicated to the case study, to better explain its characteristics.
The authors present a state-of-the-art of “Social Sustainability Assessment” and “NAIADE Methodology”. is suggested that the methodology approach proposed by the authors be presented later in an independent section (Methods), prior to its application to the case study: Collegno. This section should be clear, so that the method can be replicated by another researcher in another case study.
- Thank you for this suggestion; as abovementioned said, we included the description of the methodology we used in a separate paragraph, to make description of both the method and the case study more understandable.
Discussion of paper is poor and the analysis should be presented with more description. The main contribution of the authors in comparison with previous works should discuss more profoundly and highlights in detail. The authors should explain clearly, what the contribution of this paper is. It should not
be a mere application of an already known methodology applied to a case study.
- Thank you for this comment. We improved the part dedicated to the discussion and we also highlighted in the conclusion the main contribution of this paper, in terms of literature review and results of the application.
Conclusions. The paragraph shall be independent from the main text without citing references. It should be concise including the most important new findings. What are the limitations of the proposed method?
- Thank you for this remark. We better underlined in the text that the main limitations of this work that are represented by the modality to obtain the social evaluation. Moreover, we also specified that future researches will be focused on this specific topic, in order to provide a more rigorous way to get these recommendations.
Other issues:
What information is obtained from each of the skateboarders should be provided. How are defined the matrices (impact and equity)?
- Thank you for this remark. In the section dedicated to the stakeholder analysis performed for this specific case study, we specified which are the information obtained and why these information are useful for our evaluation.
Please, avoid lumping references as in [1-5], [26-29] and all other. Instead, summarize the main contribution of each referenced paper in a separate sentence.
- Thank you for this suggestion, we very appreciate it. However, we decided to maintain these typology of references because within this work we aim also to give a robust literature review about the discussed topic. Thus, it is possible for the readers to have different references related to the same topic, in order to make them able to select the best one referred to their interests and works.
The language should be revised as grammatical errors are pervasive: there are typos, wrong figure captions...
- Thank you for this suggestion, we carefully revised the language and the caption.

Reviewer 3 Report
Authors present a very interesting research related to a multi-methodological approach to analyze social sustainability in relation to urban regeneration program. They underline, in a pertinent way, the complexity of the topic and its multidisciplinary. The paper is well written and structured, the literature review is wide, pertinent and complete. The stakeholder analysis permits to define the 'who', 'why' and 'how' related to the right stakeholder involvement for each developing scenario. The NAIADE method permits to objectify social and urban analysis removing the subjective aspects of each decision makers.
Major reviews:
- all the criteria (table 4) are quantitative except number 18 ("via De Angelis regeneration"): please clarify if there is not any conflicts in the methodology application;
- Chapter 2 is very long (even too much), especially paragraph 2.4. It does not add so many significant element to the main research. Please revise;
- Section 5 is too short and figure 6 can be better explained, please revise.
Minor reviews:
- in line 328 verify 'table 4';
- line 329 the categories are 6 and not 5;
- line 408 verify figure 6 caption.
Author Response
Letter to Reviewer #3
Thank you a lot for appreciating our work. We very appreciate your comments that were useful to improve our work.
Moreover, we would like to underline that the general structure of the paper has been changed, in order to respond to all the suggestions received. However, the content of the paper is the same, considering that has been accepted by all the reviewer.
The authors.
Major reviews
All the criteria (table 4) are quantitative except number 18 ("via De Angelis regeneration"): please clarify if there is not any conflicts in the methodology application;
- Thank you for this remark. We added a specific phrase in text, at line 333. Moreover, SMCE is a typology of MCA, so it is possible to consider both quantitative and qualitative criteria for the evaluation.
Chapter 2 is very long (even too much), especially paragraph 2.4. It does not add so many significant element to the main research.
- Thank you for this comment. We simplified the paragraph 2 in general and specifically we reduced the quantity of information related to the paragraph 2.4. We appreciated very much this suggestion as it was useful to focus only on the most important elements to underline in our literature review.
Please revise; - Section 5 is too short and figure 6 can be better explained, please revise.
- Thank you for this suggestion. As requested, we modified section 5, in order to improve the dissertation. Moreover, we added some new details and information related to Fig. 6.

Round 2
Reviewer 2 Report
I appreciate the authors' efforts to improve the paper. My comments have been addressed according to the suggestions.